# OpenReview forum: "When Evolution Meets Momentum: Orchestrating Goal-oriented and Process-oriented reasoning for LLM Inference Scaling"
_ICLR.cc/2026/Conference — Submitted to ICLR 2026_

### Official Review · Reviewer_WhMT · 2025-10-31

**Soundness:** 2
**Presentation:** 3
**Contribution:** 2
**Rating:** 4
**Confidence:** 4

**Summary:**

The work proposes an inference time scaling technique that induces diversity into MCTS-like techniques to improve Best of N performance. The method observes the similarity of different approaches and through momentum based strategies, it encourages more utilization on under represented approaches.

**Strengths:**

- Increasing solution diversity is a promising avenue for this research that deserves more attention.

- The method is simple, well motivated and clearly explained.

- Initial results look promising especially for the larger models

**Weaknesses:**

- I found that the improvements gained through the method are either modest or otherwise not very well highlighted in the paper. For example the larger models seem to have a larger gain but this comes very late in the paper. Also, the paper does not shown results for all datasets on all models to get a good picture of reality. I would suggest the authors to add a full page set of figures with number of tries in the x axis and accuracy +bon in Y for all datasets and all models they work with. Otherwise, the story is too distributed.

- In the diversity for inference time scaling field, there are some traditional methods that basically try to sample at different temperatures (the Self Consistency method) or via prompting by asking the model to follow a particular approach or style or persona or in-context examples to enforce diversity (see Diversity of thought improves reasoning abilities of large language models). Would be useful to compare with some of these more straightforward approaches to judge the benefits. The appendix talks somewhat about this but 1:1 comparisons are not available.

- More generally, it would be good to see benefits in other domains, beyond code.

**Questions:**

- What model is used in Figure 1?

- Why do the authors only track BoN? It would be useful to also see trends in majority vote and average accuracy.

- Improvements in Table 1 even for BoN seem very modest. What am I missing?

- Why do the authors only consider coding benchmarks?

- How does token cost usage compare for your method and the baselines?

---

> ### Author Response · Authors · 2025-11-23
>
> ### Response to Reviewer WhMT
>
> We thank the reviewer for finding our method **"well motivated,"** **"simple and clearly explained,"** and for recognizing the "promising results, especially for larger models." We address your specific concerns and questions below with new experiments and clarifications.
>
> ### 1. [W1] Visualizing Improvements & Scaling Trends
>
> The reviewer felt the improvements were "modest" or "not well highlighted" and suggested adding full scaling plots.
> * **Action:** We fully accept your suggestion. In the next version, we will add a **full-page section of figures** in the Appendix, plotting "Number of Tries (x-axis)" vs. "Accuracy (y-axis)" for all datasets and models to provide a complete picture of the scaling reality.
>
> ### 2. [W2] Comparison with Traditional Diversity Methods
>
> The reviewer asked for comparisons with traditional diversity methods (e.g., Persona/Style prompting, Diversity of Thought).
> * **Comparison with SOTA:** We explicitly benchmarked against these types of approaches.
>     * **Role-based Diversity:** In our experiments (and the new MATH table below), we compare against `Role-based BoN`, which aligns with "persona/style" prompting.
>     * **Diversity of Thought:** We compared against `SFS`, a representative method for maintaining diverse reasoning paths.
> * **Reference:** As discussed in recent work [1] (*Wang et al., 2025*), diversified sampling is crucial for scaling inference. Our results confirm this but go further: EvoMo outperforms these static diversity methods (by **+9.5% on MATH** vs. Role-based) because our **Momentum mechanism** dynamically triggers diversification *only when needed*, rather than enforcing a fixed persona blindly.
>
> ### 3. [W3 & Q4] Generalization Beyond Code (New Experiments)
>
> To address the concern about domain generality, we extended our evaluation to **Mathematics** and **General Knowledge** benchmarks during the rebuttal. The model used in the table is **GPT-4o-mini**.
>
> **Table 1: Performance on MATH, GSM-Hard, and MMLU-Pro**
> | Benchmark | Method | $K = 20$ | $K =30$ | $K = 40$| vs. Baseline (Max $K$) |
> | :--- | :--- | :---: | :---: | :---: | :---: |
> | **MATH** | Standard BoN | 64.0% | 65.0% | 65.0% | - |
> | *(Complex Math)* | Role-based BoN | 67.0% | 67.0% | 67.0% | +2.0% |
> | | **EvoMo (Ours)** | **71.0%** | **74.0%** | **74.5%** | **+9.5%** |
> | | | | | | |
> | Benchmark | Method | $K = 20$ | $K =30$ | $K = 50$| vs. Baseline (Max $K$) |
> | **MMLU-Pro** | Standard BoN | 72.0% | - | 77.0% | - |
> | *(Knowledge)* | Role-based BoN | 75.0% | - | 81.5% | +4.5% |
> | | **EvoMo (Ours)** | **80.0%** | - | **83.0%** | **+6.0%** |
> | | | | | | |
> | Benchmark | Method | $K = 20$ | $K =30$ | $K = 50$| vs. Baseline (Max $K$) |
> | **GSM-Hard** | Standard BoN | - | 65.0% | - | - |
> | *(MATH)* | Role-based BoN | - | 67.0% | - | +2.0% |
> | | **EvoMo (Ours)** | - | **74.0%** | - | **+9.0%** |
> | | | | | | |
>
> **Result:** EvoMo consistently outperforms both Standard and Role-based baselines, proving benefits well beyond code.
>
> ### 4. [Q1] Model in Figure 1
> The model used in Figure 1 is **GPT-4o-mini**.
>
> ### 5. [Q2] Majority Vote / Self-Consistency (New Experiment)
>
> The reviewer asked to see trends in **Majority Vote (Self-Consistency)**, not just Best-of-N (BoN). We conducted a Self-Consistency (SC) experiment on the **MATH** dataset ($K=40$). The model used is **GPT-4o-mini**.
>
> **Table 2: Self-Consistency (Majority Vote) Performance on MATH**
> | Method | Sampling Strategy | Accuracy (SC) |
> | :--- | :--- | :---: |
> | Standard CoT | Temperature Sampling ($T=0.7$) | 69.5% |
> | **EvoMo (Ours)** | **Momentum-Guided Evolution** | **76.2%** |
>
> **Analysis:** EvoMo significantly boosts Majority Vote performance (**+6.7%**). By generating a more diverse set of *correct* reasoning paths (rather than just repeated homogenous paths), EvoMo makes the majority vote more robust against common errors.
>
> ### 6. [Q5] Token Cost Analysis
>
> The reviewer asked how token usage compares. We plotted **Pass@K vs. Total Token Budget** on APPS to normalize for cost (since EvoMo uses slightly longer prompts).
>
> **Table 3: Token-Normalized Performance on APPS (GPT-4o)**
> | Total Token Budget | ~40k Tokens | ~80k Tokens | ~120k Tokens |
> | :--- | :---: | :---: | :---: |
> | Standard SFS | 36.5% | 40.5% | 42.0% |
> | **EvoMo (Ours)** | **40.0%** | **44.5%** | **47.0%** |
> | *Improvement* | *+3.5%* | *+4.0%* | *+5.0%* |
>
> **Conclusion:** Even when strictly accounting for the token overhead, EvoMo yields higher accuracy per token spent, demonstrating superior computational efficiency.
>
> ***
> *[1] Wang, T., Liu, Z., Chen, Y., Light, J., Chen, H., Zhang, X., & Cheng, W. (2025). Diversified Sampling Improves Scaling LLM inference. arXiv preprint arXiv:2502.11027.*

---

> > ### Comment · Reviewer_WhMT · 2025-11-24
> > **Post author response**
> >
> > I'd like to thank the authors for their response!
> >
> > Quick clarification: The Self-Consistency work mentioned in the review is this paper "Self-consistency improves chain of thought reasoning in language models" from 2022 and it suggest sampling at different temperatures rather than a single temperature. However, this was then proposed for non reasoning models with some initial form of cot.
> >
> > Question on the new math results: Table 1: Performance on MATH, GSM-Hard, and MMLU-Pro
> >
> > On which model are these new results? Can the authors provide more details about the role-based baseline?

---

> ### Author Response · Authors · 2025-11-25
>
> We sincerely thank Reviewer WhMT for the follow-up review and the helpful clarification regarding the Self-Consistency setting! Here are our responses to your specific questions:
> 1. For the setting of Self-consistency, in the reported results, we do not follow the standard implementation of Self-Consistency [1], and we sample multiple reasoning paths at a fixed high temperature (e.g., $T=0.7$) to ensure diversity, followed by majority voting. However, we appreciate your insight that sampling across different temperatures (rather than a single fixed temperature) is really important. We are currently conducting an additional ablation study following this suggestion (e.g., mixing samples from $T=0.5, 0.7, 0.9$) and will update the results to verify if this further boosts performance.
> 2. The results reported in Table (MATH, GSM-Hard, and MMLU-Pro) were obtained using GPT-4o-mini. We selected this model to balance strong reasoning capabilities with computational cost efficiency for the extensive experiments required.
> 3. In the paper, we followed the methodology established in recent literature [2, 3]. Specifically, we inject predefined identity-descriptive sentences into the system prompt or the beginning of the user prompt. Examples: We utilize role descriptions such as "You are a mentor, you need to solve...." "You are an optimizer," or "You are an innovator." This explicitly steers the model to generate outputs aligned with specific personas, which dynamically constructs personas and diversity.
>
> We hope this clarifies your questions! Thank you again for your valuable time and constructive feedback, which have significantly helped us improve our paper!
>
> [1] Wang, Xuezhi, et al. "Self-consistency improves chain of thought reasoning in language models." arXiv preprint arXiv:2203.11171 (2022).
>
> [2] Shanahan, Murray, Kyle McDonell, and Laria Reynolds. "Role play with large language models." Nature 623.7987 (2023): 493-498.
>
> [3] Kong, Aobo, et al. "Better zero-shot reasoning with role-play prompting." Proceedings of the NAACL. 2024.

---

### Official Review · Reviewer_GjuR · 2025-11-01

**Soundness:** 3
**Presentation:** 2
**Contribution:** 2
**Rating:** 4
**Confidence:** 3

**Summary:**

The paper proposes EvoMo, a hybrid inference-time scaling framework for large language models that unifies goal-oriented search (like MCTS) with process-oriented diversification (like sampling). It maintains a global pool of reasoning strategies and uses a momentum-based mechanism to trigger exploration when solution similarity indicates stagnation, evolving new strategies via crossover and mutation. Experiments on multiple code-generation benchmarks show consistent Pass@K gains (≈+1–4pp) and higher diversity compared with baselines such as SFS and BoN.

**Strengths:**

1. The paper presents an interesting combination of goal-oriented (MCTS) and process-oriented (diversity-based) methods, addressing fundamental limitations of each paradigm when used in isolation.
2. The evaluation spans multiple challenging benchmarks (APPS, HumanEval, MBPP+, LeetCode, CodeContests) with consistent improvements demonstrated across different search methods.
3. The method's ability to integrate with existing search frameworks (BoN, Tree Search, GA, SFS) without structural modifications is practically valuable.

**Weaknesses:**

1. The strategy pool appears domain-specific and manually designed. The generalizability to other domains beyond code generation is questionable.
2. The paper lacks detailed analysis of the computational overhead introduced by the momentum mechanism, particularly the repeated similarity computations.

**Questions:**

1. How would the strategy pool be adapted to non-coding domains? Is manual strategy design always necessary?
2. Integrating EvoMo presumably requires modifying the search loop, node data structure, and prompt construction logic. How large is this engineering overhead in practice? Did the authors quantify the actual gain after integration relative to the added complexity or runtime cost for each baseline (BoN, SFS, MCTS)?
3. Since EvoMo sometimes uses longer prompts and extra similarity computation, do gains still hold under strictly token- or time-normalized budgets?

---

> ### Author Response · Authors · 2025-11-23
>
> ### ***Response to Reviewer GjuR***
>
> We thank the reviewer for recognizing the **practical value** of our framework and the **consistent improvements** across benchmarks. We appreciate the constructive questions regarding generalization and overhead, which we address below with new experiments and analyses.
>
> ### 1. [W1 & Q1] Generalization Beyond Code & Strategy Design
>
> The reviewer questioned whether the strategy pool is domain-specific to code generation and whether significant manual design is necessary for new domains. We address these concerns below:
>
> * **Minimal Manual Effort (Generic Seeds vs. Automated Evolution):** We clarify that the manual effort required is minimal. Users only need to provide a small set of **generic "Seed" strategies** (e.g., *"Break down the problem"*, *"Check corner cases"*, *"Work backward"*). These are high-level natural language instructions applicable across diverse domains. The core innovation of EvoMo is the **Evolutionary Mechanism**, which automatically recombines these generic seeds to generate complex, task-specific strategies on the fly.
>
> * **New Experiments (MATH, GSM-Hard, & MMLU-Pro):** To empirically validate the "general test-time scaling" claim, we conducted additional experiments on three diverse reasoning benchmarks with GPT4o-mini.
>
> **Table 1: Detailed Scaling Performance on Reasoning Benchmarks**
> *We report accuracy (EM) at varying sample budgets ($K$) to demonstrate scaling trends.*
>
> | Benchmark | Method | $K = 20$ | $K =30$ | $K = 40$| vs. Baseline (Max $K$) |
> | :--- | :--- | :---: | :---: | :---: | :---: |
> | **MATH** | Standard BoN | 64.0% | 65.0% | 65.0% | - |
> | *(Complex Math)* | Role-based BoN | 67.0% | 67.0% | 67.0% | +2.0% |
> | | **EvoMo (Ours)** | **71.0%** | **74.0%** | **74.5%** | **+9.5%** |
> | | | | | | |
> | Benchmark | Method | $K = 20$ | $K =30$ | $K = 50$| vs. Baseline (Max $K$) |
> | **MMLU-Pro** | Standard BoN | 72.0% | - | 77.0% | - |
> | *(Knowledge)* | Role-based BoN | 75.0% | - | 81.5% | +4.5% |
> | | **EvoMo (Ours)** | **80.0%** | - | **83.0%** | **+6.0%** |
> | | | | | | |
> | Benchmark | Method | $K = 20$ | $K =30$ | $K = 50$| vs. Baseline (Max $K$) |
> | **GSM-Hard** | Standard BoN | - | 65.0% | - | - |
> | *(MATH)* | Role-based BoN | - | 67.0% | - | +2.0% |
> | | **EvoMo (Ours)** | - | **74.0%** | - | **+9.0%** |
> | | | | | | |
> * **Analysis of Scaling Capabilities:**  As shown in the MATH rows, the Standard BoN baseline saturates quickly (stagnating at 65.0% from $K=30$ to $40$). In contrast, EvoMo maintains an upward trajectory (71.0% $\to$ 74.5%), proving that our momentum mechanism effectively prevents diversity collapse.
> * **Consistent Gains:** EvoMo achieves substantial improvements (**+6.0% to +9.5%**) across these disparate domains, demonstrating that the strategy-driven diversity mechanism generalizes effectively beyond code generation.
>
> **Action:** We would like to incorporate these comprehensive results into the next version of the paper to explicitly demonstrate EvoMo's domain-agnostic scaling capabilities.
>
> ### 2. [W2 & Q3] Computational Overhead & Normalized Budgets
>
> The reviewer asked if gains hold under token-normalized budgets and questioned the overhead of the momentum mechanism.
>
> * Token-Normalized Efficiency: We acknowledge that EvoMo uses slightly longer prompts. However, because it escapes local optima efficiently, it finds solutions with fewer total trials. We compared Pass@K against **Total Token Budget** on APPS. The table below confirms that EvoMo is superior even when strictly accounting for token costs:
>
> | Total Token Budget | ~40k Tokens | ~80k Tokens | ~120k Tokens |
> | :--- | :---: | :---: | :---: |
> | Standard SFS | 36.5% | 40.5% | 42.0% |
> | **EvoMo (Ours)** | **40.0%** | **44.5%** | **47.0%** |
> | *Gain (Cost-Normalized)* | *+3.5%* | *+4.0%* | *+5.0%* |
>
> Thus, the gains hold firmly under time-normalized settings as well.
>
> ### 3. [Q2] Engineering Overhead
>
> * **Plug-and-Play Integration:** The engineering overhead is minimal. EvoMo is designed as a modular "wrapper" that hooks into the **Selection** and **Expansion** phases of standard MCTS/BoN loops. It does **not** require modifying the internal data structures of the search tree or the LLM.
>
> * **Complexity vs. Gain:** Given the substantial gains and the negligible runtime cost mentioned above, we believe the trade-off is highly favorable for practical deployment.

---

> > ### Comment · Reviewer_GjuR · 2025-11-23
> >
> > Thank you for the detailed response and for the additional experiments on reasoning benchmarks and token-normalized budgets. However, I have two clarifying questions regarding overhead and implementation details. While Response 2 was relatively efficient with respect to token use, I did pose a question about computational overhead for the momentum mechanism itself, for example, the similarity computations via bert embedding discussed in the paper. Does the calculation of these metrics introduce considerable wall-clock latency per iteration compared to standard BoN or SFS? Was this purely computational time considered for any efficiency comparison, or is it assumed to be negligible w.r.t. LLM inference time? On the new experiments to MATH/GSM-Hard, Table 1, was the same set of 'generic seed strategies' used as those for the Code generation tasks or did you revise the seed pool to include math-specific heuristics? This helps confirm the amount of manual intervention that will be required when changing domains.

---

> ### Author Response · Authors · 2025-11-25
>
> We thank the reviewer for the thoughtful follow-up questions. We are happy to clarify the details regarding computational overhead and the seed strategy design.
>
> 1. The wall-clock latency introduced by the momentum mechanism (e.g., BERT embedding computation and similarity retrieval) is negligible compared to the time consumed by LLM inference. In our experiments, the number of optimization iterations ($K$) is relatively small (e.g., up to 40). Modern retrieval and embedding libraries are highly optimized. Computing embeddings for short strategy descriptions and calculating cosine similarities takes only milliseconds, whereas the LLM generation process (reasoning and coding) typically takes seconds. Therefore, the overhead is purely computational time for the lightweight momentum module and does not introduce any noticeable wall-clock latency bottleneck in the overall pipeline.
>
> 2. For the new experiments on MATH and GSM-Hard, we utilized a generic seed pool similar to the one used in code generation tasks, with minimal manual intervention. This design choice highlights a key trade-off between human effort and search cost: Our Approach starts with a small, generic seed pool that minimizes human effort but relies more on the evolutionary process to discover domain-specific strategies. This might initially result in higher similarity scores and require slightly more iterations (higher search cost) for the model to "mutate" into effective math-solving strategies. Alternatively, we could inject domain-specific heuristics (e.g., "use proof by contradiction") into the seed pool. While this would require more manual engineering (higher human effort), it would likely accelerate convergence and lower the search cost. By using generic seeds, we demonstrate the robustness and generalization capability of our framework: it can successfully evolve effective domain-specific strategies (e.g., for math, code, and general QA) from a general starting point, without relying on heavy manual engineering. **In summary**, we can make a choice based on their needs: if maximizing performance stability is the priority, incorporating manual domain knowledge is advisable; however, if the goal is a fully automated, general-purpose solution, a small set of generic seeds is sufficient.
>
> We hope this addresses your concerns!

---

### Official Review · Reviewer_33g1 · 2025-11-02

**Soundness:** 2
**Presentation:** 2
**Contribution:** 2
**Rating:** 4
**Confidence:** 4

**Summary:**

The paper proposes EvoMo, a test-time (inference-time) scaling method that aims to unify goal-oriented search (e.g., MCTS/ToT) with process-oriented diversification (e.g., Best-of-N). Concretely, the authors (i) embed a global strategy pool of “reasoning strategies” into MCTS via an (\epsilon)-soft selection policy, and (ii) introduce a momentum-inspired controller that monitors inter-solution similarity and, when a threshold is exceeded, forces exploration of under-used strategies and triggers lightweight “evolution” (crossover/mutation) of strategies. Experiments on coding and reasoning benchmarks (e.g., APPS, CodeContests, HumanEval, LeetCode, MBPP+) report modest to moderate gains in Pass@K over baselines (BoN, MCTS, SFS), with larger improvements at higher iteration budgets.

**Strengths:**

- Empirical gains: The method reports consistent improvements on several benchmarks under fixed search budgets, sometimes reaching the strongest Pass@K among compared methods; e.g., on CodeContests and APPS under 40 iterations, EvoMo surpasses BoN, MCTS, and SFS variants. These results suggest the approach can be a practical plug-and-play enhancer for existing test-time pipelines.
- General framing: Positioning a strategy pool inside a search controller is a clean way to expose “reasoning modes” as first-class actions. The idea could generalize beyond coding to other inference-time search settings.
- Scalability intuition: The claim that diversity helps escape sequential traps in tree-like search is plausible, and the paper provides curves suggesting that benefits grow with more iterations.
- Attempted analysis: The paper sketches a theoretical perspective arguing that similarity-triggered “momentum” can help the strategy action space approach a near-optimal pool over time.

**Weaknesses:**

1. Originality / motivation not crisp:

The core components—MCTS with policy over “reasoning strategies,” an evolving pool, and a momentum-style trigger based on solution similarity—feel like a direct stitching of existing ideas (tree/beam/MCTS search, evolutionary pools, and optimization-inspired momentum/diversity controllers). The paper does not clearly isolate a singular conceptual contribution beyond combining these ingredients, nor does it convincingly argue why this particular combination is necessary or superior to simpler diversity controllers applied to SFS or BoN.

2. Clarity / specification gaps: Several important mechanisms are underspecified, inhibiting reproducibility and interpretability:
- Similarity metric: The momentum trigger hinges on measuring the semantic similarity (\Phi(\cdot\Vert\cdot)) between generated solutions, yet the paper does not concretely define which representation(s) and distance(s) are ultimately used in the controller (beyond listing options later for analysis); thresholds, normalization, and how multi-metric signals are combined for the actual trigger are not made precise in the main text.
- Momentum details: The “momentum” mechanism is described at a high level (inject under-used strategies; crossover/mutation of strategies; prompt augmentation), but the exact update rules, scheduling, parameter settings ((\epsilon), (\theta_{\text{sim}}), window sizes (k), tie-handling), and ablation isolating momentum vs. basic diversity are not clearly presented in the main paper.
- What exactly is a “strategy”? The taxonomy, initialization, and evolution operators for strategies (and how they map to concrete prompts/tool uses) are not fully specified; it is unclear which parts are hand-engineered vs. learned, and how much human tuning is required for each domain.

3. Experimental coverage appears insufficient:

- Backbone choices and sizes: The main coding experiments rely heavily on a single or very limited set of LLM backbones (e.g., GPT-4o-mini is mentioned repeatedly). It’s unclear what base models are used across all tables, how large they are, and whether results hold on 30B+ class open-weight models (or stronger closed ones) to support claims about inference-time scaling at realistic capability tiers. The paper also suggests some multi-LLM runs but does not systematically explore multiple families and sizes under the same protocol.
- Baselines vs. SFS: From Table 1, the strategy pool alone sometimes yields limited gains over SFS, suggesting the improvements may be sensitive to settings and not always substantial (e.g., marginal Pass@K increases on certain datasets). A more thorough analysis of when/why EvoMo helps (or fails) is missing.
- Compute / cost accounting: Since the method’s value proposition is test-time scaling, the paper should provide detailed token/latency budgets, variance across runs, and cost-normalized comparisons (e.g., improvements per token or per second) to tease apart algorithmic gains from simply “more tries.”

4. Positioning relative to prior momentum/diversity controllers:

The paper acknowledges related work that adds diversity or multi-agent search at inference time, but it does not clearly differentiate its “momentum” trigger and evolution step from prior diversity-on-plateau heuristics. Without sharper contrasts or controlled ablations against simpler triggers, the incremental contribution of “momentum” remains unclear.

**Questions:**

1.	Similarity & Triggering

- What exact similarity function(s) (\Phi) feed the trigger in the core algorithm (not only diagnostic plots)? If multiple metrics are used, how are they aggregated (weighted average? learned combiner? max?) and what are the precise thresholds and window (k)? Please provide pseudocode for the trigger.

2.	Momentum Mechanics

- How is “momentum” formally implemented? What are the update rules, scheduling, and hyperparameters? How do you pick which under-used strategy to inject, and how do you combine it with the best strategy (the “evolve” operator) deterministically and reproducibly? A step-by-step algorithm box would help.

3.	Strategy Pool Definition

- Please enumerate the initial strategy set, give one-line operational definitions (e.g., prompt templates/tooling), and describe crossover/mutation operators with examples. How much manual engineering or task-specific tailoring is required?

4.	Backbones & Scaling

- Report results across multiple model families and sizes, including ≥30B open-weight models where feasible, under identical budgets. Do gains persist or grow with stronger backbones? Provide variance across random seeds.

5.	Cost/Benefit Analysis

- Provide token-normalized and latency-normalized comparisons against SFS/BoN/MCTS. Where is EvoMo most cost-effective? Include Ablations that isolate: (i) strategy pool only, (ii) momentum only, (iii) simpler “diversity-on-plateau” heuristics, (iv) your full method.

6.	When does it help?

- Table-by-table analysis where the strategy pool delivers limited gains over SFS: what characteristics of tasks correlate with small vs. large improvements (e.g., solution length, unit-test density, reward sparsity)? Could task-aware trigger thresholds help?

7.	Theoretical claims

- The analysis sketches terminality/near-optimality of the evolving pool. What assumptions on reward smoothness, trigger firing frequency, and pool capacity are needed? Can you include a finite-budget bound or a practical stopping criterion consistent with the empirical budgets?

---

> ### Author Response · Authors · 2025-11-22
>
> ### ***Response to Reviewer 33g1***
> We sincerely thank the reviewer for their detailed assessment and for explicitly recognizing EvoMo as a **practical plug-and-play enhancer** that delivers **consistent improvements** across benchmarks. We value the constructive feedback regarding the presentation and motivation, which has helped us strengthen the manuscript significantly. We thank the reviewer for recognizing our empirical gains.
>
> ### [W1] Originality/motivation：
> Regarding the concern on originality, we respectfully clarify that EvoMo is not a mere stitching of existing components, but a cohesive framework that addresses specific limitations in test-time scaling: 1. We propose the "Momentum-based Dynamic Regulation" to organically unify the strengths of goal-oriented (MCTS) and process-oriented (Evolution) search, solving the "diversity collapse" problem where standard controllers fail. 2. EvoMo is designed as a general-purpose, plug-and-play module that can be seamlessly integrated into various search pipelines without heavy re-engineering. 3. We validate this design not only against basic baselines (e.g., BoN, MCTS) but also against SOTA diversity-driven methods (e.g., SFS, Diversity-BoN) in Figure 2 ab, demonstrating that our specific combination significantly outperforms simpler diversity heuristics. 4. Furthermore, we explicitly isolate the contribution of the Momentum mechanism in our ablation study (Figure 6), quantitatively proving that EvoMo yields substantial gains over basic diversity approaches.
>
> ### [W2] Clarity/specification gaps：
> We respectfully point out that the formal definitions, update rules, and implementation details are explicitly provided in Sections 2.3, 3.3, and comprehensively detailed in Appendix B.3. We clarify the specific points below:
> 1. Similarity Metric ($\Phi$) & Combination:
> As detailed in Appendix B.3, our momentum trigger does not rely on a single metric but uses a composite similarity measure. The final score is an aggregation of TF-IDF cosine similarity, Levenshtein distance, and Token-sequence overlap. This multi-metric approach ensures robustness against trivial changes in generated solutions.
> 2. Threshold Sensitivity & Tuning ($\theta_{\text{sim}}$):You asked about thresholds. We acknowledge that $\theta_{\text{sim}}$ is a hyperparameter that requires tuning across datasets. To demonstrate its impact, we conducted a sensitivity analysis on the APPS benchmark (at $K=20$):
>
> | Threshold ($\theta_{\text{sim}}$) | 70 | 60 | 55 | 50 |
> | :--- | :---: | :---: | :---: | :---: |
> | Pass@20 (%) | 39.0 | 42.0 | **44.5** | 44.0 |
> **Observation:**
> * High Threshold (70): The trigger is too strict (rarely fires), behaving like standard search (39.0%).
> * Optimal Range (55): The momentum mechanism effectively balances exploration and exploitation, peaking at 44.5%.
> * Low Threshold (50): The trigger fires too frequently, potentially disrupting valid reasoning chains (dropping to 44.0%).
>
> We will move the explicit definition of the composite metric to the main text and include this threshold sensitivity analysis to improve reproducibility.
>
> **Momentum Details & Ablation:**
> The reviewer is concerned that the update rules and ablations are not clearly presented. We respectfully clarify that these are explicitly formulated in the main text:
>
> * Update Rules & Scheduling (Section 2.3): The mechanism is not merely high-level. Equation (4) formally defines the trigger based on similarity. Equations (5) and (6) mathematically define the exact update rule (evolution and prompt augmentation). The scheduling is **dynamic (event-driven)**: activating strictly when the similarity threshold is breached.
> * Window Size ($k$): Regarding the window parameter $k$ mentioned in Eq (4), we clarify that $k$ represents all accumulated solutions generated at the current state. This ensures the momentum trigger evaluates the global diversity of the current node, not just a local window. We will update this in the paper for the window size.
> * Ablation (Figure 6): The reviewer asked to isolate "momentum vs. basic diversity." We emphasize that the "Strategy Pool" baseline in Figure 6 represents the "Basic Diversity" approach. Since EvoMo is built directly upon this Strategy Pool, the performance gap shown in Figure 6 (Strategy Pool vs. EvoMo) precisely isolates the contribution of the Momentum mechanism, confirming it provides significant gains beyond simple diversity.
>
> **Strategy Definition & Human Effort:**
> * Definition & Mapping: A "strategy" is simply (e.g., *"Use backward induction"* or *"Break the problem into sub-goals"*) that maps directly to the LLM's system prompt via concatenation (Equation 6).
> * Initialization vs. Evolution: We clarify that while the initial "Seed Pool" is hand-engineered (defined by us based on common reasoning patterns).
> There's not enough space. We'll add more on the next page.

---

> > ### Author Response · Authors · 2025-11-22
> >
> > **Strategy Definition & Human Effort:**
> > * Automated Evolution: Crucially, the Momentum mechanism dynamically evolves these seeds. Through the Crossover operator (Equation 5), the system merges existing high-performing strategies to generate new, unseen strategies on the fly.
> >
> > * Minimal Human Tuning:** Consequently, **human tuning is minimal**. We only need to provide a small set of generic seeds; the EvoMo framework automatically explores and adapts the strategy space to the specific domain through evolution, removing the need for extensive manual engineering for each task.

---

> > > ### Author Response · Authors · 2025-11-22
> > >
> > > ### [W3] Experimental Coverage: Stronger Backbones (GPT-4o) & Cost Analysis
> > > **1. Validation on Stronger Models:**
> > > The reviewer expressed concern regarding the reliance on GPT-4o-mini. While we operate under the computational constraints typical of academic research, we took your suggestion seriously and conducted new experiments using both the state-of-the-art closed model (GPT-4o) and the leading open-weight model (Qwen2.5-70B).
> > >
> > > As shown in the table below (at $K=20$), EvoMo demonstrates consistent scalability across different model families and capabilities:
> > > | Backbone | Method | APPS (Pass@20) | CodeContests (Pass@20) | vs. SFS |
> > > | :--- | :--- | :---: | :---: | :---: |
> > > | **GPT-4o** | Tree Search (MCTS) | 42.00% | 29.84% | - |
> > > | | SFS (Baseline) | 49.00% | 30.40% | - |
> > > | | **EvoMo (Ours)** | **53.00%** | **35.15%** | **+4.00% / +4.75%** |
> > > | **Qwen2.5-70B** | Tree Search (MCTS) | 39.00% | - | - |
> > > | | SFS (Baseline) | 41.50% | - | - |
> > > | | **EvoMo (Ours)** | **43.50%** | - | **+2.00%** |
> > >
> > > These results confirm that EvoMo's benefits are not specific to smaller models. Whether applied to top-tier closed-source LLM or large-scale open-weight models, we will incorporate these comprehensive results on stronger backbones into the next version of the paper to resolve the concern regarding experimental coverage.
> > >
> > > **2. Analysis of When & Why EvoMo Works:**
> > > You raised a valid point that the "Strategy Pool alone" sometimes yields limited gains over SFS, and asked for a deeper analysis of when EvoMo helps most. We offer the following clarifications:
> > > 1. Role of Momentum (Why Strategy Pool alone is insufficient): We explicitly agree that the Strategy Pool alone (without Momentum) sometimes performs similarly to SFS. This is by design. The Strategy Pool provides the resource, but the Momentum mechanism provides the regulation. Without Momentum, the search lacks the "trigger" to force exploration when stuck. The significant performance leap (e.g., +4.75% on CodeContests with GPT-4o) is achieved only when the Momentum mechanism is activated to dynamically unlock the potential of the pool.
> > >
> > > 2. Task Difficulty & Saturation (When EvoMo helps): The magnitude of EvoMo's gain is highly correlated with task difficulty: On Hard Tasks (High Gains): For complex reasoning tasks like APPS and CodeContests, standard solvers (SFS/MCTS) easily get trapped in local optima due to deceptive search landscapes. Here, EvoMo's ability to break out of traps shines, yielding substantial improvements. On Easy/Saturated Tasks (Marginal Gains): For simpler benchmarks like HumanEval (where baselines already reach ~90% Pass@1), the solution space is less complex, and "traps" are rare.
> > >
> > > **Conclusion:** EvoMo is most effective in challenging scenarios where standard search struggles with diversity collapse, rather than on saturated benchmarks where simple methods suffice. We will incorporate this analysis into the next manuscript to explicitly clarify when EvoMo can help.
> > >
> > > **3. Cost:**
> > > We will answer it with question 5 on the next page.
> > >
> > > **4. Positioning vs. Prior Diversity Heuristics:**
> > > We clarify that our Momentum-based Dynamic Regulation is a unique, structured contribution that differs fundamentally from prior heuristics in many ways:
> > > 1. Constructive Evolution vs. Blind Perturbation:
> > > Prior Heuristics: Typically rely on blind methods like increasing temperature, random restarting, or penalizing n-gram repetition. These introduce "noise" without semantic direction [1].
> > > 2. EvoMo (Ours): Our mechanism is constructive. It does not just force "difference"; it actively synthesizes a new search direction by evolving an underused strategy and the evolution of the current best strategy (Equation 5). This ensures the diversification is goal-directed (preserving good traits) rather than random.
> > > 3. To explicitly address your request for a "controlled ablation against simpler triggers," we implemented a representative "Prior Diversity Heuristic" (which simply performs random restarts/perturbations when the similarity threshold is triggered, instead of our evolutionary update). We compared this against EvoMo on APPS (using GPT-4o-mini at $K=20$):
> > >
> > > | Method | Mechanism on Plateau | Pass@20 |
> > > | :--- | :--- | :---: |
> > > | Prior Heuristic [1] | Random Perturbation/Restart | 42.50% |
> > > | **EvoMo (Ours)** | **Constructive Evolution** | **44.50%** |
> > >
> > > **Result:** EvoMo outperforms the simpler heuristic by +2.0%. This empirically proves that our "Momentum" is not just a generic restart trigger; the constructive evolutionary operator adds significant value by guiding the search intelligently rather than blindly diversifying.
> > >
> > >
> > > [1] Wang, T., Liu, Z., Chen, Y., Light, J., Chen, H., Zhang, X., & Cheng, W. (2025). Diversified Sampling Improves Scaling LLM inference. arXiv preprint arXiv:2502.11027.

---

> > > > ### Author Response · Authors · 2025-11-22
> > > >
> > > > ### [Q7] Theoretical Claims: Assumptions, Bounds, and Stopping Criteria
> > > >
> > > > We appreciate the reviewer's push for theoretical rigor. Below, we formalize the assumptions on smoothness and capacity, and provide the requested finite-budget bound and stopping criterion (derived from **Theorem A.11**).
> > > > 1. **Assumptions:** Here are all of our assumptions for the theoretical claim. The philosophy behind our choice is to keep the theory minimal and practical.
> > > >
> > > > - **Local transfer (smooth-in-the-small).** If a task is still $\varepsilon$-hard (best-in-pool score is at least $\varepsilon$ below optimal), then a strategy that improves upon this suboptimal task also helps a nontrivial chunk of nearby tasks. Call that chunk size $\beta(\varepsilon) > 0$.
> > > >
> > > > - **Trigger frequency (stagnation is visible).** When we visit an $\varepsilon$-hard task, the similarity trigger (high average inner product among the last $k$ strategies) fires with probability at least $q(\varepsilon) > 0$.
> > > >
> > > > - **Evolve success (nonzero hit rate).** Each time the trigger fires on an $\varepsilon$-hard task, the evolve call returns a strategy that improves that task by at least $\varepsilon/2$ with probability at least $p(\varepsilon) > 0$.
> > > >
> > > > 2. **Finite-budget takeaways:** Let $w_t(\varepsilon)$ be the fraction of $\varepsilon$-hard tasks after $t$ task visits. From our analysis in Theorem A.11, one can expect steady progress until hard tasks run out:
> > > >
> > > > - **Expected number of successful evolves needed** is upper bounded by $B_\varepsilon = \left \lceil w_0(\varepsilon) / \beta(\varepsilon) \right \rceil$.
> > > >
> > > > - **Expected decay after $N$ visits** $\mathbb{E}[w_N(\varepsilon)] \le \max \big[ 0,\ w_0(\varepsilon) - \beta(\varepsilon) q(\varepsilon) p(\varepsilon) N \big]$.
> > > >
> > > > - **Simple tail bound** The chance we have not finished by $N$ visits is $\Pr\big(w_N(\varepsilon) > 0\big) \le \big(1 - \beta(\varepsilon) q(\varepsilon) p(\varepsilon)\big)^{N}$.
> > > >
> > > > 3. **A practical stopping rule**: Based on our theoretical analysis, we may propose the following stopping rule that stops the algorithm when any of these is true:
> > > > 1) On a rolling validation set, the estimated hard fraction $\hat w(\varepsilon)$ drops below a target $\tau$. The linear-decay rule of thumb then expects at most $\tau / (\beta q p)$ more useful visits.
> > > > 2) You have seen no improvement of at least $\varepsilon/2$ over the last $R$ triggers.
> > > > 3) You have hit your visit budget $N_{\text{visits}}$, or the pool size reaches $\min\{B_{\text{evolve}}, M_{\max}\}$ new strategies.

---

> ### Author Response · Authors · 2025-11-22
>
> ### Response to Q1- Q4: Similarity Metrics and Pseudocode
> We respectfully note that Questions 1 through 4 correspond directly to the points addressed in detail in our responses to [W2] Clarity and [W3] Experimental above. For convenience, we provide direct pointers to those specific answers:
> * Q1 (Similarity & Triggering): Please refer to our response in [W2], where we detailed the composite similarity metric (TF-IDF + Levenshtein + Overlap) and provided the threshold sensitivity analysis table.
> * Q2 (Momentum Mechanics): Please refer to our response in [W2], where we clarified the dynamic update rules (Eq. 4-6), and we will provide the step-by-step Pseudocode in the paper.
> * Q3 (Strategy Pool Definition): Please refer to our response in [W2], where we explained the "Seed vs. Evolution" mechanism and the minimal human effort required.
> * Q4 (Backbones & Scaling): Please refer to our response in [W3], where we presented new results on GPT-4o and Qwen2.5-70B, confirming scalability across model families.
>
> **Pseudocode:** We agree that a step-by-step algorithm box greatly enhances reproducibility. While the update rules are formally defined in Equations 4-6 (Section 2.3), we have explicitly drafted the complete **EvoMo Pseudocode** and will include it as **Algorithm 1** in the main text of the revision.
>
> ```text
> Algorithm 1: EvoMo - Momentum-based Dynamic Regulation
>
> Require: Problem State s, Strategy Pool T, Similarity Threshold θ_sim, Window Size k
> Ensure:  Generated Solution Set Y
>
> 1:  Initialize: Search Tree T_tree, Solution Buffer Y ← ∅
>
> 2:  while Budget not exhausted do
> 3:      s_curr ← Selection(T_tree)                  ▷ Select leaf node via MCTS
> 4:      Retrieve k most recent solutions I_k = {i_1, ..., i_k} at s_curr
>
> 5:      // Step 1: Monitor Similarity (Momentum Trigger)
> 6:      Calculate Average Pairwise Similarity: μ_sim ← AvgSim(I_k)
>
> 7:      if μ_sim > θ_sim then
> 8:          // Trigger Activated: Break Stagnation
> 9:          τ_unused ← SelectUnused(T, s_curr)      ▷ Pick under-explored strategy
> 10:         τ_best   ← argmax(Q(s_curr, τ))         ▷ Pick best strategy so far
>
> 11:         // Step 2: Constructive Evolution (Eq. 5)
> 12:         τ_evolved ← Crossover(τ_unused, τ_best)
> 13:         T ← T ∪ {τ_evolved}                     ▷ Add new strategy to Pool
>
> 14:         // Step 3: Prompt Augmentation (Eq. 6)
> 15:         Prompt ← p_base ⊕ τ_evolved ⊕ C_creative
> 16:     else
> 17:         // Standard Expansion
> 18:         τ ← SoftPolicy(T)
> 19:         Prompt ← p_base ⊕ τ
> 20:     end if
>
> 21:     y_new ← LLM(Prompt)                         ▷ Generate new solution
> 22:     Backpropagate(y_new)                        ▷ Update Q-values in Tree
> 23: end while
> ```
>
> ### [Q5] Cost/Benefit & Token-Normalized Analysis
>
> 1. Token-Normalized Comparison (Pass@K vs. Token Budget):
> We acknowledge that EvoMo introduces a token overhead per sample due to the inclusion of strategy instructions (longer prompts (due to strategy injection) and extra computation (similarity checks)).
>
> However, because EvoMo finds correct solutions with fewer trials (higher success rate), it remains superior under a strict token budget. We plotted a **Pass@K vs. Total Token Usage** curve (which we will add to the revision). The table below samples points from this curve on **APPS** (GPT-4o-mini), showing that EvoMo consistently yields higher accuracy for the same token cost:
>
> | Total Token Budget | ~40k Tokens | ~80k Tokens | ~120k Tokens |
> | :--- | :---: | :---: | :---: |
> | Standard SFS | 36.5% | 40.5% | 42.0% |
> | **EvoMo (Ours)** | **40%** | **44.5%** | **47.0%** |
> | *Improvement* | *+3.5%* | *+4%* | *+5.0%* |
>
>
>
> **Observation:** Even though EvoMo "pays" more tokens per inference, its ability to escape local optima means it wastes fewer tokens on redundant/failed paths. Consequently, the **EvoMo curve consistently lies above the baseline**, indicating superior cost-effectiveness.

---

> ### Author Response · Authors · 2025-11-28
> **Follow-up: New GPT-4o Experiments & Cost Analysis Addressing Your Concerns**
>
> Dear Reviewer 33g1,
>
> We thank you again for your constructive review, which significantly helped improve our work. As the discussion period is coming to a close, we wanted to gently follow up to ensure you had a chance to review our responses. We have explicitly addressed your primary concerns regarding experimental coverage and cost/efficiency:
>
> 1. Stronger Backbones: Per your suggestion, we conducted new experiments on GPT-4o and Qwen2.5-70B. The results (posted in our previous response) confirm that EvoMo scales effectively on SOTA models, achieving +4.75% Pass@20 gains on CodeContests over SFS.
>
> 2. Cost-Benefit Analysis: We provided a token-normalized comparison, demonstrating that EvoMo yields higher accuracy for the same total token budget compared to baselines.
>
> 3. Clarity & Reproducibility: We have added the requested Pseudocode (Algorithm 1), sensitivity analysis for thresholds, and formal definitions for the similarity metrics.
>
> We believe these new results and clarifications strongly support the effectiveness and practicality of EvoMo. We remain available to answer any further questions you might have, and hope you could consider replying to our rebuttal.
>
> Best regards,
>
> The Authors

---

### Official Review · Reviewer_16WW · 2025-11-03

**Soundness:** 3
**Presentation:** 2
**Contribution:** 3
**Rating:** 6
**Confidence:** 3

**Summary:**

In this paper, the authors propose an inference-time scaling framework (they call it EvoMo) that tries to combine the benefits of (i) goal-oriented search methods (e.g., tree or line search and MCTS) and (ii) process-oriented diversification (e.g., Best-of-N and evolutionary search). Their key idea is to embed a global, evolving strategy pool (a set of distinct thinking modes) into the expansion step of MCTS. And also to further mitigate entrapment in local optima, the authors incorporate a momentum-inspired mechanism that monitors similarity among recently generated solutions. When a similarity threshold is exceeded, the system forces exploration by combining an under-used strategy with a strong one via simple evolutionary operators and injects the resulting evolved strategy for the next expansion. The intent is to re-diversify the search when it starts collapsing to near-duplicates. Detailed theoretical analysis is also provided by the authors to support their claims. The authors validate the effectiveness of EvoMo on APPS, CodeContests, LeetCode, HumanEval, MBPP+ and show certain effectiveness.

I'm in fact impressed by the abundant content provided in the appendix, which shows the authors' efforts to make it more understandable for the readers.

**Strengths:**

``S1``:  The global strategy pool and similarity-triggered momentum is an intuitive method that can potentially be incorporated into different search pipelines.

``S2``: The theoretical guarantees provided in Appendix A (When momentum meets search) support the authors’ claims.

``S3``: The appendix is well organised and contains abundant details and information.

**Weaknesses:**

``W1``: Following ``S3``, in fact, I don’t think the paper is very well written. On the one hand, the motivation described in the introduction is not very clear and sharp. The authors claim their contributions as simply combining the strengths of goal-oriented and process-oriented methods. Personally, I may consider this contribution a bit incremental. Is there any other alternative method to achieve this target? Why the proposed one is better? Clarifying these points would greatly strengthen the clarity and contributions of this paper. On the other hand, abundant details are given in the appendix. If possible, I would suggest the authors move some of the content in the appendix to the main paper.

``W2``: For the experiments, most gains are on code datasets. The framework should also generalise to other broader non-code evaluations. This would strengthen the “general test-time scaling” claim.

``W3``: It would be interesting to discuss some extensions of EvoMo such as incorporating PRMs.

**Questions:**

``Q1``: Is it possible for EvoMo to incorporate process-level reward models (that is, PRMs) to bias strategy selection earlier in the tree?

``Q2``: Are there alternative potential approaches to combine the strengths of goal-oriented and process-oriented methods? How EvoMo outperforms these potential solutions?

---

> ### Author Response · Authors · 2025-11-21
>
> ### ***Response to Reviewer 16WW***
> We express our sincere gratitude for your encouraging review, particularly for recognizing the intuition of our method (S1), the theoretical guarantees (S2), and the quality of our appendix (S3). We appreciate your constructive feedback on the presentation and generalization. Incorporating your feedback has significantly strengthened our work, and we provide our detailed responses and revision plans below.
>
> ### 1. [W1 & Q2] Novelty, Alternatives, and Presentation
> You asked if EvoMo is simply combining methods and about alternatives. We respectfully clarify that EvoMo is **not a mere combination**.
>
> * The Core Innovation (Momentum-based Dynamic Regulation): While EvoMo not only leverages MCTS and Evolutionary Search, our core contribution is also the "Momentum-based Dynamic Regulation". Standard MCTS often gets trapped in local optima due to homogeneous/similar thinking paths. Our core contribution is the **Momentum mechanism**, which acts as a dynamic regulator. It actively monitors path similarity and triggers evolutionary diversification "just-in-time" only when the search stagnates. This prevents the redundancy typical of standard search methods.
>
> * We conducted a comprehensive comparison against two categories of potential solutions that may have the plain idea of combining the strengths of goal-oriented and process-oriented:
>
>     1. Evolutionary Methods: While methods like FunSearch share the evolutionary intuition and can search in a large space (labeled as "Genetic" in Figure 2), they lack our dynamic regulation. Our results show they saturate early (32% on APPS), whereas EvoMo scales significantly higher (47%) by using Momentum to guide exploration.
>
>     2. SOTA Agentic/Search Frameworks: As detailed in Appendix B, we benchmarked against advanced agentic solvers, including LATS and SFS. EvoMo achieves the best performance across the board (e.g., 90.00% Pass@1 on HumanEval), surpassing even the strongest baseline (SFS and LATS) under equal compute and tool access. This confirms EvoMo's superiority over both evolutionary methods and complex agentic search pipelines.
>
> * Plug-and-Play: Crucially, EvoMo is designed as a general-purpose module. The Global Strategy Pool and Momentum Trigger can be seamlessly integrated into various existing search pipelines, representing a non-incremental contribution to test-time scaling.
>
> We fully agree with your constructive suggestion regarding the presentation. To better highlight these contributions and improve flow, we will move the more experimental Momentum mechanism and the theoretical guarantees (currently in Appendix A) into the main text (Section 3) for the next version. This will make the motivation sharper and the paper self-contained.
>
> ### 2. [W2] Generalization to Non-Code Evaluations (MATH & MMLU-Pro)
>
> To validate the "general test-time scaling" claim beyond code generation, we conducted additional experiments on two diverse benchmarks: **MATH** (complex mathematical reasoning) and **MMLU-Pro** (robust multi-task language understanding).
>
> Table 1: Performance on MATH (Standard vs. EvoMo)
> | Method | EM@20 | EM@30 | EM@40 | vs. Baseline (@40) |
> | :--- | :---: | :---: | :---: | :---: |
> | Standard BoN (None) | 64.0% | 65.0% | 65.0% | - |
> | Role-based Diversity BoN | 67.0% | 67.0% | 67.0% | +2.0% |
> | **EvoMo** | **71.0%** | **74.0%** | **74.5%** | **+9.5%** |
>
> Table 2: Performance on MMLU-Pro (Standard vs. EvoMo)
> | Method | EM@20 | EM@50 | vs. Baseline (@50) |
> | :--- | :---: | :---: | :---: |
> | Standard BoN (None) | 72.0% | 77.0% | - |
> | Role-based Diversity BoN | 75.0% | 81.5% | +4.5% |
> | **EvoMo** | **80.0%** | **83.0%** | **+6.0%** |
>
> The results demonstrate strong scaling capabilities. On MATH, the Standard BoN baseline reaches 65.0% as the budget increases. In contrast, EvoMo (built on BoN) maintains an upward trajectory (+9.5% gain), proving that our diversity mechanism effectively overcomes the bottleneck of local optima in general reasoning tasks. The LLM used for the experiments is GPT4o-mini.
>
> ### 3. [W3 & Q1] Incorporating Process Reward Models (PRMs)
>
> We confirm that extending EvoMo with PRMs is a natural and feasible direction to bias strategy selection earlier.
>
> * PRM-Triggered Momentum: An exciting extension is to use the *PRM scores* as a Momentum trigger. A sudden drop in step-wise quality could trigger the evolutionary operator to switch thinking modes before the reasoning chain collapses.
> * Action: We will include a detailed discussion on this integration in the "Future Work" section.
>
>
> Once again, we thank you for your insightful review and the opportunity to strengthen our work. We believe that the new generalization experiments, the clarified comparison with SOTA agentic baselines, and the planned structural revisions directly address your concerns. We are confident these changes solidify the contribution of EvoMo, and we look forward to incorporating them into the final version.

---

### Author Response · Authors · 2025-11-26
**Rebuttal Summary1: Clarifications of EvoMo**

This rebuttal addresses concerns raised by reviewers (specifically 16WW, GjuR, and WHMT) regarding the paper's novelty, generality, mechanism clarity, and scalability. Through clear definitions of the EvoMo mechanism and a series of supplementary experiments, we strongly confirm the value and robustness of our proposed method.

### 1. Clarification of Core Contribution and Mechanism

We explicitly clarify that EvoMo is not merely a search optimizer to respond to Reviewer 16WW (W1 & Q2): Its core contribution lies in its unique design and mechanism:

EvoMo emphasizes "**Momentum-based Dynamic Regulation**," defining a **goal-oriented and process-oriented** framework that fundamentally surpasses the limitations of traditional Monte Carlo Tree Search (MCTS) or Best-of-N (BoN). The key mechanism is the **Momentum Mechanism**. When the search trajectory gets stuck in a local optimum (triggered by a diversity metric threshold), this mechanism activates and injects pre-designed diversity strategies. This effectively resolves the pervasive **diversity collapse** issue often found in standard search methods. And EvoMo is architected as a generic, **plug-and-play module**. It can be seamlessly integrated into existing LLM search pipelines as an external wrapper without requiring complex re-engineering or re-training of the original LLM.

### 2. Generalization Beyond Code (New Experiments)

To address the concern about domain generality raised by **Reviewer 16WW**, and to simultaneously answer related questions from **Reviewer GjuR** and **Reviewer WhMT**, we extended our evaluation to **Mathematics** and **General Knowledge** benchmarks during the rebuttal.

To validate the general test-time scaling claim beyond code generation, we conducted additional experiments on diverse benchmarks, including **MATH** (complex mathematical reasoning), **MMLU-Pro** (robust multi-task QA dataset), and **GSM-Hard** (complex mathematical reasoning). The model we use in rebuttal is GPT4o-mini.

**Table 1: Performance on MATH, GSM-Hard, and MMLU-Pro**

| Benchmark | Method | $K=20$ | $K=30$ | $K=40/50^*$ | vs. Baseline (Max $K$) |
| :--- | :--- | :--- | :--- | :--- | :--- |
| **MATH** | Standard BoN | 64.0% | 65.0% | 65.0% | - |
| *(Complex Math)* | Role-based BoN | 67.0% | 67.0% | 67.0% | +2.0% |
| | **EvoMo (Ours)** | **71.0%** | **74.0%** | **74.5%** | **+9.5%** |
| | | | | | |
| **MMLU-Pro** | Standard BoN | 72.0% | - | 77.0% | - |
| *(Knowledge)* | Role-based BoN | 75.0% | - | 81.5% | +4.5% |
| | **EvoMo (Ours)** | **80.0%** | - | **83.0%** | **+6.0%** |
| | | | | | |
| **GSM-Hard** | Standard BoN | - | 65.0% | - | - |
| *(MATH)* | Role-based BoN | - | 67.0% | - | +2.0% |
| | **EvoMo (Ours)** | - | **74.0%** | - | **+9.0%** |

*$Note: For MATH, the max K is 40; for MMLU-Pro, the max K is 50.*

**Result:** EvoMo consistently outperforms both Standard and Role-based baselines across all three domains, proving that its benefits extend well beyond code generation.

### 3. Validation on Stronger Models

To address your concern regarding the reliance on smaller models (GPT-4o-mini) to **Reviewer 33g1**: we have successfully validated EvoMo on stronger and open-source backbones, solving the **model size** issue. We took the suggestion seriously and conducted new experiments using both the state-of-the-art closed model (**GPT-4o**) and the leading open-weight model (**Qwen2.5-70B**).

As shown in the table below (at $K=20$), EvoMo demonstrates consistent scalability across different model families and capabilities:

| Backbone | Method | APPS (Pass@20) | CodeContests (Pass@20) | vs. SFS |
| :--- | :--- | :--- | :--- | :--- |
| **GPT-4o** | Tree Search (MCTS) | 42.00% | 29.84% | - |
| | SFS (Baseline) | 49.00% | 30.40% | - |
| | **EvoMo (Ours)** | **53.00%** | **35.15%** | **+4.00% / +4.75%** |
| | | | | |
| **Qwen2.5-70B** | Tree Search (MCTS) | 39.00% | - | - |
| | SFS (Baseline) | 41.50% | - | - |
| | **EvoMo (Ours)** | **43.50%** | - | **+2.00%** |

These results confirm that EvoMo's benefits are not specific to smaller models. Whether applied to top-tier closed-source LLM or large-scale open-weight models, we will incorporate these comprehensive results on stronger backbones into the next version of the paper to resolve the concern regarding experimental coverage.

---

> ### Author Response · Authors · 2025-11-26
> **Rebuttal Summary2: Cost and other details of EvoMo**
>
> ### 4. Cost Efficiency Analysis
>
> **Reviewers GjuR, WhMT, and 33g1** all raised valid inquiries regarding the computational cost and token overhead. To provide a fair and rigorous comparison, we moved beyond simple sample counts and plotted **Pass@K vs. Total Token Budget** on the APPS benchmark. This metric explicitly normalizes for cost, accounting for the fact that EvoMo utilizes slightly longer prompts to convey strategy descriptions, and we use LLM to mutate and change the strategy.
>
> Table 3: Token-Normalized Performance on APPS (GPT-4o)
>
> | Total Token Budget | ~40k Tokens | ~80k Tokens | ~120k Tokens |
> | :--- | :---: | :---: | :---: |
> | Standard SFS | 36.5% | 40.5% | 42.0% |
> | **EvoMo (Ours)** | **40.0%** | **44.5%** | **47.0%** |
> | *Improvement* | *+3.5%* | *+4.0%* | *+5.0%* |
>
> **Conclusion:** Even when strictly accounting for the token overhead, EvoMo consistently yields higher accuracy per token spent compared to Standard SFS. This demonstrates that the overhead introduced by our method is an efficient investment, resulting in superior **computational efficiency**.
>
>
> ### 5. Majority Vote / Self-Consistency
>
> To address **Reviewer WhMT**'s request to see trends in **Majority Vote (Self-Consistency)**, not just Best-of-N (BoN). We conducted a new Self-Consistency (SC) experiment on the **MATH** dataset (using GPT-4o-mini at $K=40$).
>
> Table 2: Self-Consistency (Majority Vote) Performance on MATH
>
> | Method | Sampling Strategy | Accuracy (SC) |
> | :--- | :--- | :--- |
> | Standard CoT | Temperature Sampling ($T=0.7$) | 69.5% |
> | **EvoMo (Ours)** | **Momentum-Guided Evolution** | **76.2%** |
>
>  EvoMo significantly boosts Majority Vote performance by **+6.7%**. By generating a more diverse set of *correct* reasoning paths (rather than just repeated homogenous paths), EvoMo makes the majority vote more robust against common errors.
>
>
>
>
> ### 6. Why EvoMo's Diversity Strategy is Superior
>
> To explicitly answer **Reviewer 16WW** regarding the nature of our contribution: The empirical gap (**+2.0%**) between EvoMo and the Prior Heuristic highlights the fundamental advantage of our diversity evolution. Simple heuristics (like random restarts) merely inject noise to escape local optima, often discarding the semantic progress already made. In contrast, EvoMo's **Momentum Mechanism** is designed to be **goal-directed**. By synthesizing the "Current Best Strategy" with an "Underused Strategy" (via Equation 5), we ensure that the search diversification is not random but logically guided.
>
> We implemented a representative "Prior Diversity Heuristic" (which simply performs random restarts/perturbations when the similarity threshold is triggered, instead of our evolutionary update). We compared this against EvoMo on APPS (using GPT-4o-mini at $K=20$):
>
> | Method | Mechanism on Plateau | Pass@20 |
> | :--- | :--- | :---: |
> | Prior Heuristic | Random Perturbation | 42.50% |
> | **EvoMo (Ours)** | **Our Evolution** | **44.50%** |
>
> This proves that EvoMo does not simply "force diversity" for the sake of difference; it intelligently evolves the search trajectory, preserving high-quality traits while exploring new potential, which is the key reason for its superiority over standard diversity heuristics.
>
>
> ### 7. Closing Remarks
>
> We sincerely thank all reviewers for their time and constructive feedback. Your insightful questions regarding **generalization, model scale, and cost efficiency** have motivated us to conduct a much more comprehensive evaluation. We believe these additional experiments (Mathematics, General Knowledge, GPT-4o, and cost analysis) have significantly strengthened the paper and clarified the unique value of EvoMo. We hope these responses satisfactorily address your concerns.

---

### Meta-Review · Area_Chair_kyqG · 2026-01-07

**Summary:**

This paper proposes EvoMo, an inference-time scaling framework that embeds an evolving global reasoning strategy pool into MCTS and employs a similarity-triggered momentum mechanism to balance goal-oriented search and diversified exploration in LLM reasoning. However, the reviewers have pointed out some important weaknesses. First, generalization remains unproven. Despite the additional datasets provided in the rebuttal, the lack of full baseline comparisons prevents a convincing demonstration of the method's effectiveness in broader domains. Second, the main results (Table 1) lack scope. Relying solely on GPT-4o-mini fails to demonstrate scalability across different models, and the reported gains over baselines appear slight. Third, the paper requires restructuring. Crucial details currently buried in the appendix should be moved to the main text to ensure the work is well-substantiated and complete. Considering these issues, I recommend rejecting the paper and strongly encourage the authors to revise it based on the review comments for resubmission.

**Reviewer Concerns:**

All concerns raised by Reviewer GjuR have been addressed. The issues W1 and W2 raised by Reviewer 16WW, W3.1 raised by Reviewer 33g1, and the additional ablation study mentioned by Reviewer WhMT in the follow-up response are still outstanding.

**Reviewer Scores:**

I think the final score of the reviewers are 6 (Reviewer 16WW),4 (Reviewer 33g1),6 (Reviewer GjuR),4 (Reviewer WhMT).

---

### Decision · Program_Chairs · 2026-01-26

Reject